# Proteomics in the World of Induced Pluripotent Stem Cells

**DOI:** 10.3390/cells8070703

**Published:** 2019-07-11

**Authors:** Rafael Soares Lindoso, Tais H. Kasai-Brunswick, Gustavo Monnerat Cahli, Federica Collino, Adriana Bastos Carvalho, Antonio Carlos Campos de Carvalho, Adalberto Vieyra

**Affiliations:** 1Carlos Chagas Filho Institute of Biophysics and National Center for Structural Biology and Bioimaging/CENABIO, Federal University of Rio de Janeiro, Rio de Janeiro 21941-102, Brazil; 2Laboratory of Proteomics, LADETEC, Institute of Chemistry, Federal University of Rio de Janeiro, Rio de Janeiro 21941-598, Brazil; 3Department of Biomedical Sciences, University of Padova, 35131 Padua, Italy; 4Graduate Program in Translational Biomedicine, Grande Rio University, Duque de Caxias 25071-202, Brazil

**Keywords:** induced pluripotent stem cells, proteomics, embryonic stem cells, differentiation, organoids

## Abstract

Omics approaches have significantly impacted knowledge about molecular signaling pathways driving cell function. Induced pluripotent stem cells (iPSC) have revolutionized the field of biological sciences and proteomics and, in particular, has been instrumental in identifying key elements operating during the maintenance of the pluripotent state and the differentiation process to the diverse cell types that form organisms. This review covers the evolution of conceptual and methodological strategies in proteomics; briefly describes the generation of iPSC from a historical perspective, the state-of-the-art of iPSC-based proteomics; and compares data on the proteome and transcriptome of iPSC to that of embryonic stem cells (ESC). Finally, proteomics of healthy and diseased cells and organoids differentiated from iPSC are analyzed.

## 1. Introduction: A historical View and Evolution of Conceptual and Methodological Strategies in Proteomics

Proteomics is a large-scale protein analysis that includes the identification, quantification, and posttranslational modification, among other relevant information regarding proteins in a tissue, cell, or biofluid. The proteome is the entire set of the proteins produced or modified by an organism or system. The term proteome was first described in 1994 by Marc Wilkins. Proteomics is based in ground-breaking innovation in bioinformatics and liquid chromatography associated with mass spectrometry and, currently, is passing through numerous recent innovations in all these points [1,2]. Among the other “omics” analysis, proteomics is one of the most broadly used for the investigation of induced pluripotent stem cells (iPSC), as shown in Figure 1.

Due to the vast complexity of the mammalian proteome, pre-analytical strategies have been historically applied to improve protein identification by mass spectrometry. In this regard, the first fractionation strategy used in samples such as cells and tissues was based on electrophoresis initially, one-dimensional electrophoresis followed by two-dimensional fractionation. In the electrophoresis gel, each dot represents one or more proteins that can be further selected and prepared for a mass spectrometry analysis. Due to the multidimensional structure of the proteins and the extensive amino acid sequences, several steps, such as alkylation, reduction, and digestion, had to be optimized to improve protein identification by mass spectrometry-based proteomics. Since each dot of the gel had to be individually picked, electrophoresis-based proteomics was very time-consuming, limiting the number of proteins and samples analyzed in a given study [3,4]. In this context, the problem of protein fractionation was partially solved due to advances in liquid chromatography associated with mass spectrometry, mainly regarding the application of nanoflow separation chromatography associated with mass spectrometry (nLC-MS) [5,6].

It is essential to highlight that the digestion step during sample preparation is of crucial importance for bottom-up proteomics. Several methodologies have been used to improve peptide preparation for LC-MS analysis. Protein digestion is usually conducted with one or with a combination of enzymes or chemicals to generate the peptides of interest. Since available enzymes provide distinct cleavage sites in the proteins, researchers can select the enzyme type based on the cleavage to peptides of interest for the particular experiments, for example, of longer or shorter peptide lengths. The most broadly used enzyme is trypsin, which cleaves peptide chains at the carboxyl side of the amino acids lysine or arginine, except when followed by proline. Therefore, by bioinformatics, it is straightforward to predict the peptides generated from a given protein after digestion. The enzyme Lys-C is also widely used and can be applied in combination with trypsin to achieve better digestion efficiency. As mentioned, different protocols are used, and these include the following: (i) differences in enzyme concentration—normally varying from 1:50 or 1:100 (enzyme to protein concentration) for trypsin—(ii) differences in duration of incubation—12 to 18 h are broadly used—and (iii) differences in temperature and pH of the reaction, which are also essential to achieve higher digestion performances. During data analysis for protein identification, the researcher can select the number of missed cleavages, since it is known that the digestion efficiency is not 100%. There are online tools to predict potential cleavage sites by proteases or chemicals in a given protein sequence, such as the PeptideCutter with more than 30 enzymes and chemical options (https://web.expasy.org/peptide_cutter/).

An additional and crucial aspect of recent advances in the proteomics field is related to bioinformatic tools. An nLC-MS experiment is typically conducted using a 1 to 3 h gradient protocol. Therefore, thousands of mass spectra are generated, creating a massive amount of information demanding bioinformatic algorithms for protein identification, such as Mascot [7] and SEQUEST [8] search engines. Several open softwares such as Pattern-Lab [9] and MaxQuant [10] have been publicly available for the scientific community, including all tools necessary for non-targeted nLC-MS-based proteomics analysis. Regarding proteomic databases, the protein sequence database for several organisms is freely available for download from online platforms such as UniProt and SwissProt [11].

Besides protein identification in the initial qualitative LC-MS based analysis, an essential topic in proteomics nowadays is related to quantitative strategies. To allow quantification, 2 major methods have been developed: (i) label-free quantification and (ii) isotopically labeled quantification. The first method does not require any additional step during sample preparation; therefore, it is the most broadly used approach. In label-free quantification, the bioinformatic algorithms use the areas of the peptide precursors in the mass spectra, as well as spectra counting for peptide/protein quantification. However, the results generated by the label-free protocols leads to relative quantification, where the researcher can estimate the fold change variation between samples, preferably analyzed as one batch, to avoid analytical fluctuations [12]. The isotopically labeled strategy, on the other hand, uses known isotopes for peptide labeling during sample preparation (i.e., Isobaric tags for relative and absolute quantification (iTRAQ) and tandem mass tag (TMT)) [13] or during cell culture using labeled amino acids in the growth medium (Stable Isotope Labeling by/with Amino acids in Cell culture (SILAC) [14]). Since different known isotopes are added to the experimental group samples, by the mass-to-charge ratio differences, one can identify the labeling and can perform quantification using the peptide areas or the intensity of the reporting ion. Consequently, additional tools are needed for data analysis, such as Pattern-Lab and PEAK. However, the labeling reagents are generally costly, and labeling demands additional sample preparation steps and more sophisticated informatics skills.

The gold standard method for absolute protein quantification is conducted using high purity heavy labeled peptides (such as absolute quantification (AQUA) peptides from SIGMA-ALDRICH) in a targeted proteomics approach. In the targeted analysis, the precursor ions, both the spiked heavy peptide standard and the light endogenous peptide, are monitored, followed by fragmentation and monitoring of the product ions. In target proteomics analysis, quantification can be conducted in either a lower resolution mass spectrometer (such as a triple quadrupole) using selected reaction monitoring (SRM) protocols, providing a less expensive tool, or in a high resolution mass spectrometer (such as the Orbitrap analyzer) using parallel reaction monitoring (PRM), in which several fragments of the same precursor ion are analyzed in high resolution. Since the modern mass spectrometers are very fast, different groups have created innovative methods described as a semi-targeted approach [15,16,17], in which hundreds of peptides are monitored in a single experiment.

In addition to the presence and abundance of a given protein in a cell, a key mechanistic information is related to its function. Therefore, the investigation of posttranslational modifications (PTMs) of proteins in stem cells, as well as in differentiated cells, is crucial for understanding the physiological and pathological roles of the protein. Since PTMs promote known mass values modifications in a given peptide, alterations in the mass-to-charge ratio can be used to identify the presence of different PTMs. Several studies have described novel methods to evaluate the presence of PTMs in mammalian cells, such as phosphorylation, glycosylation, acetylation, and ubiquitination. In this case, additional steps are required during sample preparation, aiming to enrich the proteins presenting the PTMs of interest before the mass spectrometry analysis. In addition, the collision cell used for peptide fragmentation must be adapted as well as the bioinformatic strategies [18,19]. Moreover, the subcellular localization of a given protein is also of interest in stem cell biology, mainly for network investigation and protein function understanding, and innovative proteomics approaches can also perform that. Thus, recent publications have reported protein-specific localization, either by subcellular enrichment by antibody-based assay or bar-coding using cell fractionation strategies for a higher throughput investigation [20].

Recent advances in proteomics have been achieved, creating novel opportunities to further improve the identification of protein isoforms in mammalian iPSC. In the field of top-down proteomics, no digestion step is required and, therefore, the protein can be analyzed in its natural form (native proteomics) or its full sequence. In this context, recent publications described several isoforms of the same protein in a biofluid or cellular extract. This is of crucial importance for stem cell research since it is known that protein isoforms might perform different biological roles and that the isoform profile of specific cells can vary among species and during pathological processes. However, due to the technological limitations of Fourier transform-based mass spectrometry, proteins with higher masses cannot be analyzed by top-down proteomics. In this regard, better protein ionization must be achieved, as well as improvements in protein fragmentation in the collision cell of the mass spectrometer for the MS2 or MSn spectra generation, associated with new bioinformatics tools and databases [21].

## 2. Induced Pluripotent Stem Cells (iPSC): A Historical Perspective

Induced pluripotent stem cells (iPSC) were described 13 years ago [22], but the scientific foundations that culminated in their discovery have been more than a century in the making. It started in the late 1800s when August Weismann developed his barrier theory. According to him, germ cells generate somatic cells but genetic information cannot be passed from the latter to the former. Hence, the barrier concept established the paradigm that somatic cells could not produce a new organism [23]. Conrad Waddington’s epigenetic landscape also contributed to this paradigm by establishing that cell differentiation through embryonic development was comparable to marbles rolling down a hill in which the ridges and valleys represented different cells’ fates. Importantly, the marbles cannot go up the hill, illustrating that differentiated cells could not generate new organisms [23].

Since paradigms exist to be broken, John Gurdon proved in 1962 that a somatic cell could generate a new organism [24]. He produced typically developing embryos using nuclei obtained from intestinal cells inserted into enucleated eggs, a process called somatic nuclear cell transfer (SNCT) [25]. This was reproduced for the first time in mammals with Dolly [26] and later in ESC [27]. Gurdon’s discovery delineated 2 critical concepts. The first was that somatic cells do possess all the genetic information necessary to generate a new organism. The second was that unknown molecules present in the cytoplasm of the egg enabled the reprogramming of the somatic nucleus.

From 1962 to 2006, at least 2 other advances largely contributed to the discovery of iPSC: The isolation of ESC in mice [28] and humans [29] defined culture conditions to maintain pluripotency in vitro. The same requirements would later be applied to iPSC. Last but not least, the description of master transcription factors in 1987 [30] gave us a hint as to which type of unknown molecules, capable of modifying cell fate, were present in the egg’s cytoplasm.

Building on the work of several other scientists that had described transcription factors necessary for the maintenance of pluripotency in embryos and ESC, Takahashi and Yamanaka selected and forced the exogenous expression of 24 candidate transcription factors in mouse embryonic fibroblasts, culminating in the discovery of the Yamanaka factors and mouse iPSC [22].

Shortly after, iPSCs were generated from human cells [31,32] and a new field of research was launched in both basic and translational science. Laboratories started to investigate the biological mechanisms of the reprogramming process, and several models were proposed to explain it [33,34,35,36]. In addition, the same rationale used to select the Yamanaka transcription factors was applied to the field of direct reprogramming, in which cells are converted from one differentiated phenotype to another without going through the pluripotent state [37,38,39]. In parallel to the basic research advances, many laboratories invested in iPSC applications. The most obvious one is regenerative medicine, which required methodological modifications to the reprogramming process, namely the generation of integration-free [40,41,42,43] and xeno-free [44,45] iPSC. In 2014, a patient with macular degeneration underwent autologous transplantation with iPSC-derived retinal pigment epithelial cells [46]. Although the procedure was considered safe, the generation of autologous iPSC for every patient was deemed to be infeasible due to cost and time constraints [47]. To solve this issue, iPSC haplobanks are being created in different countries, and the first patient to receive allogeneic Human Leukocyte Antigen (HLA)-matched iPSC was transplanted in 2017 [48].

Other important iPSC applications are disease modeling, drug discovery, and toxicity tests. Since iPSC maintain patient genetic information and can be differentiated into several cell types, they are an ideal resource to investigate genetic mechanisms of disease in vitro [49,50,51,52,53,54] and to screen for new or off-label uses of drugs. Moreover, cardiomyocytes and hepatocytes derived from iPSC can be used to test drug toxicity before they come to the market.

iPSCs have revolutionized the field of biological sciences, but there is still much to be learned about these fascinating cells. In this review, we will explore the world of iPSC proteomics in the next 2 sections.

## 3. Proteomics of iPSC

As pointed out above, the generation of iPSC derived from somatic cells has opened a new path for regenerative medicine and represents an invaluable tool for studies of human development, monogenic disease modeling, and drug tests [55,56,57]. Right after the consolidation of iPSC generation by different labs, the first scientific question concerned the biological similarities and differences that iPSC shared with embryonic stem cells (ESC). Aspects such as genetic stability, epigenomic, and transcriptome profiles have been extensively investigated by comparing iPSC and ESC to characterize a “pluripotent identity” [58,59]. These processes, however, do not reflect the final levels of the cell’s molecular effectors: its proteins. In this context, proteomics emerged as a new approach that could establish differences on the regulation of expressions at the posttranslational levels, providing additional clues to the understanding of the biological importance of molecular similarities or differences between these pluripotent stem cells.

Independent groups conducted proteomic screenings comparing iPSC and ESC, showing that these cell types present remarkable similarities at the proteome level [60,61,62,63,64]. Phanstiel et al. were the first to compare the proteome and phosphoproteome of 4 human iPSC and 4 human ESC lineages in biological triplicate by mass spectrometry-based proteomics using iTRAQ. The peptides were fractioned by strong cation exchange (SCX), and the phosphopeptides were enriched by immobilized metal affinity chromatography followed by high-resolution LC-MS/MS analysis. The transcriptome of these cells was also analyzed by RNA sequencing [61].

This exciting work showed that iPSC and ESC were almost identical at the proteome level and that discrete differences between iPSC and ESC were only detectable when biological replicates were added, increasing the sample size, and with it, statistical power. For example, the analysis of 4 iPSC lines and 4 ESC lines (8 cell lines) revealed only 1 transcript, 5 proteins, and 4 phosphorylation sites that were statistically different between these 2 pluripotent cell types. The inclusion of 2 more replicates from each cell type (24 samples in total) increased these numbers to 1560 transcripts, 293 proteins, and 292 phosphorylation sites, which were differentially expressed between ESC and iPSC. This concordance is most impressive, taking into account that the cells were obtained from different donors and considering that more than 90% of these molecules differed by less than two-fold (Figure 2; [61]). The analysis showed that the biological processes which were enriched in iPSC when compared to ESC were related to mesodermal lineage cellular functions (muscle system process, muscle contraction, and wound healing) characteristic of the fibroblast lineage used to generate the iPSC [61]. These results were supported by data obtained from several other groups that suggested that iPSCs retain some residual epigenetic marks (an epigenetic memory) of the cells from which they were derived [58,65,66].

Munoz et al. achieved similar conclusions regarding proteome similarities between iPSC and ESC, showing that 97.8% of the “*confidently quantified proteins*” are identical in both cell lines. The proteome relative quantification was conducted in 1 ESC, 2 iPSC, and their precursor fibroblast cell lines. Peptides were labeled using triplex dimethyl chemistry, equally mixed and pre-fractionated by using strong cation exchange (SCX), which separates peptides according to their charged state. They were then sequenced using targeted fragmentation schemes. This methodology enhanced peptide identification, allowing the researchers to identify 10,628 proteins. Only 58 proteins were differentially expressed in iPSC when compared to ESC. Out of these proteins, 46 were upregulated in ESC and were related to antigen processing and metabolism of amino acids. The other 12 proteins were increased in iPSC and were related to cell-adhesion and ectoderm and mesoderm development (Figure 3; [60]).

Benevento and Munoz compared the work of both groups to evaluate if the different approaches generated similar results. A comparison of the total number of proteins identified by these two laboratories showed a low overlap: 3180 proteins were common to both datasets, while 3581 proteins were only identified by the Phanstiel et al. study and 7578 proteins were only to be found in the work by Munoz et al. Differences in methodologies (quantification methods, type of database search algorithms, and statistical criteria) could explain the discrepancies in the results. This was, in fact, demonstrated when the Phanstiel et al. data were reanalyzed using the same parameters as Munoz et al. Using the same strategy, the overlap in identified proteins added 3646 extra proteins to the intersection of the 2 proteomes. Only three upregulated proteins that were found in ESC when compared to iPSC (CRABP1, AK3, and SLC2A1) were common to both proteome groups while no downregulated proteins appeared at the intersection [60,61,67].

The combination of the proteome with transcriptome analysis has been used to investigate mechanisms of gene expression regulation. Phanstiel et al. could not find correspondence between RNA sequencing studies and proteome results. Additionally, when they compared their differentially expressed protein list with transcriptome data obtained from independent groups, they found that the proteins were also not coded by the differentially expressed genes [61]. In contrast, Munoz et al. showed that some of the differentially expressed proteins in iPSC presented compatible changes in mRNA. Despite this, several other genes did not exhibit a similar correlation, indicating the need to conduct more studies combining transcriptome–proteome analyses [60].

Kim et al. also compared the proteome of one ESC line, one iPSC line derived from human newborn foreskin fibroblasts (hFFs), and hFFs themselves. The protein lysates were separated by 2-D gel electrophoresis and identified and classified by LC-MS/MS. The authors also reported that iPSC and ESC are almost identical at the protein level, but evaluation of the differences found between the pluripotent cells and hFFS could add insights about the reprogramming process. As an example, the heterochromatin protein 1-β (HP1β) was upregulated in iPSC and ESC when compared to donor cells, and its biological function was related to chromatin remodeling. Proteins related to glycolytic enzymes (GAPDH, phosphoglycerate kinase 1, triosephosphate isomerase 1, and lactate dehydrogenase B) were differentially expressed in iPSC and ESC when compared to hFFs, suggesting that glycolytic metabolism is the primary energy generation system in pluripotent stem cells. The nucleoporin p54 (Nup54) was lower in iPSC and ESC when compared to hFFs, suggesting that the composition of the nuclear pore complex was crucial in the reprogramming process. The increased levels of the protein SET in ESC and iPSC could also play a role in the reprogramming process, considering that the overexpression of SET is related to gene silencing [62,68].

Following the same rationale, Faradonbeh et al. compared two ESC lineages with seven iPSC lines obtained from different genetic backgrounds (2 from a healthy individual, 3 from a normal individual with Bombay blood group phenotype, and 2 from a patient with tyrosinemia). They found only 48 different proteins between ESC and iPSC. Comparing these studies, just one protein appeared in both lists (GLRX3) [62,69].

This lack of reproducible results reinforced the importance of analyzing iPSC from different genetic backgrounds generated in the same way submitted to the same methodological quantitative mass spectrometry-based proteome evaluation to establish a comprehensive proteomic map of iPSC. The human Induced Pluripotent Stem Cell Initiative (HipSci) identified more than 16,000 protein groups, encoded by over 10,500 different genes by analyzing 217 iPSC lines obtained from 163 donors (healthy and disease cohorts). This large data set provides insights into the metabolism, DNA repair, and cell cycle of iPSC as well as defines primed pluripotency markers, connecting the proteome profile information with its biological function [70].

Brenes et al. showed that iPSC express high levels of key cell cycle regulators (D type cyclins, mitotic cyclins) and DNA replication complexes and low levels of CDK inhibitors, which prevent cell cycle progression. This profile is related to the high cell division rates of iPSC. In addition, due to their high proliferative capacity and potential to differentiate into cells from the three germ layers, iPSCs are more susceptible to DNA damage, enhanced rates of mutations, and cell death. Thus, in order to protect iPSC from these alterations, some proteins are highly expressed, such as inducible DNA response damage factors, RPA proteins (related to DNA homologous repair system), XRCCC6/5 (necessary at the non-homologous end joining DNA repair system), CHECK1 and CHECK2, and p53 DNA damage-induced transcription factor. The high level of glycolytic enzymes and multiple glucose transporters (GLUT1 and GLUT3) expressed by iPSC suggest that glycolysis plays an essential role in iPSC cell metabolism, as previously suggested by Kim et al. The group also created a free iPSC spectral library with the raw data and integrated the protein-level information into an open-access Encyclopedia of Proteome Dynamics available at (www.peptracker.com/epd) [62,70].

After establishing that ESC and iPSC proteomes are almost identical and revealing most of iPSC proteins’ biological functions, the efforts in the field turned to the development of more economical and faster strategies to identify pluripotency and to find putative new pluripotent markers. 

Yamana et al. developed a new approach to conduct an in-depth analysis of iPSC proteome which is faster than the traditional mass-spectrometry-based tests (<10 days vs. 30 days) and uses small amounts of sample (<100 μg vs. <1000 μg): the nanoLC-MS/MS, using meter-scale monolithic silica C18 capillary columns without sample fractionation. They compared lysates derived from 5 different iPSC with 3 different fibroblasts. In addition to categorizing the proteome profiles into 2 distinct groups—fibroblasts and iPSC-by hierarchical cluster analysis, the quantitative proteome analysis revealed that iPSCs contain more proteins related to “transcription regulation” whereas fibroblasts have more “transport related” proteins [63].

Baud et al. developed a multiplexed peptide based on multiple reaction monitoring (MRM) LC-MS/MS assay, enabling quantification of 15 pluripotency markers in only seven minutes. The authors showed that the detection of OCT-4, SOX-2, LIN28, PODXL, and CD44 by targeted proteomic-based tests identified a pluripotency signature for iPSC. To validate their technique, they compared the proteome of 14 iPSC lines with two commercial genomic tests (PluriTest and Taqman® hPSC ScoreCard™). They found a high correlation between the proteomic and transcriptomic data and demonstrated that the high throughput MRM was more cost-effective and faster than the evaluated transcriptomic tests [71,72].

Another group proposed a protein panel for iPSC pluripotency identification using global and quantitative proteome analysis [68]. They compared 2 iPSC derived from fibroblasts, 2 iPSC derived from peripheral blood CD34+ cells, their primary somatic cells, and one ESC lineage by 2 ionization techniques associated with mass spectrometry: electrospray ionization (ESI)-MS^e^ and MALDI-TOF/TOF mass spectrometry.

The iPSC and ESC proteome comparison confirmed previous results, showing only subtle differences. The proteome of iPSC obtained from different parental origins was even more similar than that of iPSC and ESC. By evaluating iPSC and parental somatic cell proteome by ESI-MS^e^, they identified 220 proteins upregulated in iPSC. Among them, 21 were previously known pluripotent markers, 12 were new candidates, and 4 were somatic cell markers. The group developed a panel using 22 of these as iPSC marker proteins. ESI-MS^e^ validated the panel and MALDI-TOF/TOF using nine different iPSC lineages obtained from different somatic origins (peripheral blood CD34+ cells, umbilical vein endothelial cells, adult fibroblast, and fetal-fibroblast), different reprogramming methods (non-integrating Sendai virus, STEMCCA-loxP lentivirus, and four non-excisable lentiviruses) and generated by different laboratories [68].

In summary, these studies revealed an extremely high similarity between ESC and iPSC at the proteome level. These small differences found between iPSC and ESC are more likely to have their origin in the epigenetic memory and experimental conditions (i.e., reprogramming method, culture conditions, different somatic backgrounds of the reprogrammed cells, sample preparation methods, sensitivity of the proteomic method used, databases used for analysis, etc.) rather than in the molecular signature [58,65,66]. Moreover, the lack of correlation between gene and protein expression levels found in some studies needs to be further examined, although protein turnover may be responsible for these differences. In conclusion, the studies revealed a high similarity in the proteome of ESC and iPSC shifting the focus to a new challenge: the proteomic studies of cells differentiated from iPSC.

## 4. Proteomics of iPSC-Derived Cells

### 4.1. iPSC Differentiation

The pluripotent potential of iPSC is represented by their capability to differentiate into cell types from the 3 embryonic germ layers: ectoderm, mesoderm, and endoderm, a property initially limited to embryonic stem cells (ESC) [22,23,31]. The development of protocols to differentiate patient-specific iPSC has been proposed lately as a tool for studies on tissue damage or regeneration. In fact, it has been demonstrated that iPSC can adequately mimic previously unrepresented human diseases, such as congestive heart failure [73] and age-related macular degeneration [43]. Initially used to study single gene mutation diseases [74,75,76], iPSC can also model complex syndromes characterized by a late onset and associated with polygenic abnormalities, including numerous neurodegenerative disorders [77,78]. If, on the one hand, patient-specific iPSC technology represents a novel platform for understanding disease mechanisms, the use of iPSC (autologous or haploidentical) provides a source for mature tissues that can be used in bioengineering technologies and treatments [79].

Cultured iPSC maintained in a 2-D culture can be induced to differentiate into specific cell types by applying an orchestrated sequence of developmental steps or by inducing the expression of specific lineage-derived transcription factors (TFs) [22]. This was demonstrated to also function across remote germ layers. By using a similar approach to that used by Takahashi and Yamanaka for iPSC generation, Vierbuchen et al. demonstrated that fibroblasts could be directly converted into neurons by the conjunctional expression of specific neural TFs [38]. Subsequently, the use of combinations of TFs to induce differentiation of fibroblasts into other cell types, such as hepatocytes and cardiomyocytes, has been developed [39,80].

The last years have been devoted to the discovery of new reprogramming molecules and the development of different approaches to improve the functional maturation of differentiated iPSC products. The use of epigenetic regulators, miRNAs, and small molecules was recently proposed, alone or in combination, to induce iPSC differentiation in multiple lineages [81]. For instance, the ectopic administration of the cardiac-specific chromatin remodeling subunit, Baf60c, in conjunction with the expression of the cardiac TFs, Gata4 and Tbx5, induced non-cardiogenic mouse mesoderm to efficiently differentiate into beating cardiomyocytes by allowing the binding of the TFs to cardiac gene promoters [82]. Yoo et al. demonstrated that the overexpression of the neuronal-specific miR-9/9* and miR-124 in combination with TFs induced human fibroblasts to partially differentiate into neuron-like cells expressing microtubule-associated protein 2 (MAP2) [83]. Moreover, cardiac lineage differentiation was achieved using the cardiac miRNAs miR-1, miR-133, miR-208, and miR-499, which directly converted fibroblasts into cardiomyocyte-like cells that expressed cardiomyocyte markers, presented sarcomeric organization, and exhibited spontaneous Ca^2+^ oscillations and mechanical contractions [84].

The presence of epigenetic barriers that act as opposing factors to the epigenetic regulators on one side and the numerous tissue targets of miRNAs on the other side [85,86] cause a substantial limitation of the efficiency of these techniques to substitute transcription-factor-mediated lineage transformation. Small molecules have also been proposed as iPSC reprogramming factors [87]. Neuronal cell fate transformation was achieved using small molecule administration in independent laboratories [88,89,90]. Also, in this case, small molecules were unable to completely substitute TF administration for most of the other lineages.

It seems clear that cell fate determination factors direct the first lineage specification but are unable to drive a complete functional maturation [91]. One major challenge that remains in the iPSC field is the development of new strategies to generate terminally differentiated cells. To identify the additional factors necessary to induce a complete functional maturation of iPSC into human hepatocytes, Du et al. [92] compared the global gene expression profile of mature human hepatocytes and immature fetal hepatocytes. The authors identified a group of TFs (PROX1, CEBPA, and ATF5) involved in hepatocyte maturation. Administration of a combination of cell fate determinants and maturation TFs induced the differentiation of iPSC into mature hepatocytes (iHeps) showing drug metabolic function. Moreover, transplantation of iHeps in Tet-uPA (urokinase-type plasminogen activator)/Rag2(−/−)/γc(−/−) mice induces up to 30% liver repopulation and secretion of human albumin [92]. The same technique has been recently applied by Viiri et al. to identify the modification of alternative transcription start site (TSS) for lncRNAs and pre-miRNAs, affecting their expression during iPSC hepatic differentiation [93].

The use of 2-D cultures has been an unproductive method for the induction of a complete differentiation of iPSC in multiple experimental settings [94,95]. Two-dimensional cultures lose cell–cell communication features and have been recently substituted by self-organizing 3-D structures mimicking tissue homeostasis and complex interactions [96]. The first pioneering studies on 3-D culturing models using ESC or iPSC were applied to neuronal organoids, making them relevant models for studies on brain development [97,98]. Interestingly, Lancaster et al. first used patient-specific iPSC carrying a mutation in the gene encoding CDK5 regulatory subunit-associated protein 2 to model microcephaly, a disorder without animal models to mimic it. Mutated iPSCs have premature neuronal differentiation with respect to control iPSC-organoids, possibly recapitulating the patient-specific microcephalic phenotype [98]. The use of 3-D modeling has also been reported for cardiac organoids. Three-dimensional cultures, together with simulated microgravity, resulted in the efficient production of differentiated cardiomyocytes with elevated viability, electrophysiological properties, and pharmacological responses [99]. Recently, the possibility to generate self-organizing 3-D human blood vessel organoids from iPSC has also been explored. These organoids contain pericytes and endothelial cells, forming capillary networks that are enclosed by a basement membrane. Moreover, when transplanted into mice and exposed to a diabetic milieu, they responded to the inflammatory milieu, developing microvascular changes found in diabetic patients [100].

### 4.2. Applications of Proteomic Approaches in the Study of Differentiated Cells Derived from iPSC

As described above, the differentiation process is a complex and orchestrated sequence of events that results in phenotypic and functional changes in the cell. Such modifications are associated not only with changes concerning gene transcription but also at the epigenetic and translational levels, resulting in alterations of the cell proteomic profile along this process [101]. Diverse proteomic approaches have been established to explore the several aspects that are associated with iPSC applications.

#### 4.2.1. Cell Differentiation and Maturation

The somatic cell reprogramming technology improved the study of the regulatory mechanisms of the differentiation process. During the process of iPSC differentiation toward a mature cell phenotype, a coordinated transition of the cell proteome must occur [101]. Along this process, several checkpoints are essential to guide the cell into a specific phenotype passing through an initial differentiation phase and a late phase associated with the maturation process [102,103,104]. Proteomic studies have helped to identify critical regulatory molecules during differentiation. Hurrell et al. conducted a proteomic time course study of iPSC differentiation and maturation into hepatocyte-like cells through LC-MS/MS analysis [105]. For the evaluation of the differentiation phase, samples were collected on days 1, 3, 5, 7, 10, 25, 30, and 35 after the beginning of the protocol. In contrast, for the maturation process, samples from days 16, 20, 24, 28, 32, 36, and 40 were chosen as representative of the late phase. Within the differentiation phase, changes in the proteomic profile on day 5 were marked by downregulation of proteins such as HELLS (lymphoid-specific helicase) and TERF1 (telomeric repeat-binding factor 1) and the upregulation of NCAM1 (neural cell adhesion molecule-1) that define the definitive formation of endoderm as described in previous studies [106,107]. In addition, several proteins associated with cell cycle progression were shown to be decreased during the differentiation process, resulting in a reduction of cell proliferation that is a requirement to achieve cell maturation. Close to the final days of the differentiation stage (25–35 days), the cells presented a more hepatocyte precursor phenotype with the consistent presence of hepatocyte-specific proteins (Figure 4; [105]).

Of particular interest was the identification of the hierarchical clustering of 16 proteins marking proliferation alongside the key element involved in G1 arrest, the cyclin-dependent kinase inhibitor 1B (CDKI1B). The identification of this cluster by Hurrel et al. opens the possibility of further investigation aiming to characterize miRNAs potentially involved in the regulation of CDKI1B during iPSC differentiation into hepatocytes. For example, the hepatomiRNoma reviewed by Bronte et al. [108] indicates the relevance of miRNA-221 during liver regeneration [109] and tumorigenesis [110]. Only a few studies focusing on the proteomics of liver regeneration—a challenge in the field of regenerative medicine—have been conducted [111]. During the maturation phase, the study could not identify the presence of the major CYP450 isoforms, which are metabolizing enzymes characteristic of mature hepatocytes [112]. These results can be attributed to the incomplete maturation of hepatocytes or to limitations of the applied proteomics. In any case, the proteomic approach used by Hurrel et al. provided relevant information about regulatory molecules important for the differentiation and maturation process [105].

One aspect of great concern in iPSC differentiation and maturation protocols is the determination of how homogeneous and similar these cells are when compared to mature cells isolated from the tissue. In order to investigate the phenotypic characteristics of the generated cells, proteomic techniques have been applied. For instance, atrial and ventricular cardiomyocytes have been generated from iPSC, and their molecular characterization was achieved by a combination of transcriptome and proteome analysis [113]. The proteomic data obtained by the SILAC technique allowed the identification and quantification of a group of 3568 proteins present in cardiomyocytes derived from iPSC (Figure 5; [113]). The comparison between atrial and ventricular iPSC-derived cardiomyocyte revealed 94 proteins highly expressed and 178 proteins significantly reduced in atrial cardiomyocytes with respect to ventricular cardiomyocytes. Such protein profiles have been previously described in adult tissue through an nLC–MS analysis that characterized different regions of the human heart [114], showing that generation of mature cardiac cells from iPSC can be successfully achieved in such a manner, permitting the studies of unique characteristics of different cell types from the same tissue. In addition, the use of this proteomic technique allows the identification of several groups of proteins which were correlated with biological and functional aspects of the cardiac cells, especially with their electrophysiological properties [113].

#### 4.2.2. Proteomics in iPSC-Derived Organoids

Organoids can be defined [115,116] as artificially generated three-dimensional collections of organ-specific cell types, which develop from stem cells (including iPSC) or organ progenitors and self-organize through cell sorting and spatially restricted lineage commitment, with a structural organization similar to the corresponding organ during development or during the genesis of diseases. The use of iPSC to establish differentiation protocols into organoids brought new perspectives in regenerative medicine, drug screening, and disease modelling [116]. Organoids present several advantages for research, as their expansion capacity with the maintenance of the genetic and physiological characteristics were found in vivo [115]. In addition, organoids have become an interesting platform for testing gene editing, with CRISPR-associated protein-9 (CRISPR/Cas9) [117], for example, for the treatment of genetic diseases. 

The recent advent of iPSC organoids helps to overcome, avoid, or minimize the fact that the degree of differentiation under standard culture conditions using immortalized cells may present some discrepancies to the physiological conditions, as demonstrated in the case of proteomic data comparing immortalized podocytes with those obtained from native tissue [118]. Using iPSC-derived kidney organoids, Hale et al. conducted an interesting transcriptional and proteomic analysis of organoid-derived glomeruli (OrgGloms) and primary podocytes isolated from these glomeruli (Figure 6; [119]). The proteomic data show that OrgGloms presented an increase in the expression of the α5(IV) chain of type IV collagen network, while α3 and α4(IV) chains were less expressed. Other components of the glomerular extracellular matrix were enriched in comparison to immortalized cells. In addition, OrgGloms presented a more mature phenotype than that observed in primary podocyte or immortalized podocyte cultures, indicating that the organoid structure allows crosstalk between different cell types in the glomeruli that are essential for the podocyte maturation process [120]. From this study [119], the advantage of proteomic studies using iPSC-derived organoids emerges.

In a different model, iPSC-derived cerebral organoids from patients carrying microduplications in chromosome 16p13.11 were generated as a model for the study of biological aspects of psychiatric and neurodevelopmental disorders. Johnstone et al. [121] generated organoids from 5 different human iPSC lines, and all of them presented layers containing photoreceptors and exhibiting the capacity of light response. One relevant aspect of this disorder is associated with reduced proliferation of neural progenitor cells (NPC) [122], and therefore, the authors [121] analyzed transcriptomic and proteomic data from iPSC-derived NPCs of healthy individuals and patients with microduplications in chromosome 16p13.11. As a result of the analysis performed by the reverse-phase proteomic array (RPPA), they observed that the reduction in NPC proliferation was associated with the modulation of NF-kappaB (NFκB) and extracellular matrix signaling, as also confirmed by RNA-seq. Immunostaining of the organoids derived from the patients presented a reduction close to 80% in the staining for NFκB p65 when compared to control organoids. Such results point to a critical mechanism regulated by NFκB and show how the combination of proteomic analysis and organoid culturing can face challenging questions related to complex mechanisms associated with neuropsychiatric and neurodevelopmental disorders.

Together with organoids, the spheroids are also 3-D cell structures that are halfway between 2-D cultures and an organ or tissue. They are clumps of cells, generated by aggregation, with little or no tissular organization [122,123,124]. They can also be obtained from patient-derived iPSC differentiated into spheroids. In a recent and very interesting study, Chen et al. [125] collected peripheral blood mononuclear cells from healthy volunteers and Alzheimer´s disease (AD) patients and converted these cells into iPSC that were then differentiated into 3-D neuro-spheroids. They carried out proteomic studies (LC-MS/MS analysis, Proteome Discoverer, UniProt) by using BPMC-derived neuro-spheroids from the 2 groups. Then, they compared neuro-spheroids protein profiles with those obtained by proteomic analysis of 3 postmortem brain regions from healthy controls and AD patients: superior frontal cortex, inferior frontal cortex, and cerebellum. They identified alterations of protein abundance and function and correlated the changes with those found in proteins from the 3 brain regions, including the critical alterations of axon proteins, and characterized the top-enriched biological processes of up- and downregulated proteins, including differentially regulated phosphorylated proteins in key signaling pathways. Overall, the study by Chen et al. demonstrated that iPSC-derived neuro-spheroids could be useful biomarkers for the progression of AD lesions.

We will end this subsection with a brief discussion of the horizons opened by recent comparative studies with the simultaneous use of spheroids and organoids. Abe et al. [126] conducted a phosphoproteomic analysis of colorectal patient-derived organoids (tissue obtained during surgical resection) and spheroids obtained from the colorectal HCT116 cells. They demonstrated that phosphosignaling is differentially activated in organoids and spheroids, thus pioneering the potential of phosphoproteomic studies, in which modifications in the tumoral environment and the tissular structural organization could allow, for example, prediction of drug sensitivity and the utilization of modifications detected in individual phosphorylation networks as biomarkers of cancer evolution. It is possible to anticipate that the use of differentiated iPSC obtained from PBMC—as in the study by Chen et al. [125]—and then cultured in appropriate and selective conditions to generate organoids and spheroids will improve, in a personalized way, the applications of iPSC-based proteomics and phosphoproteomics in the diagnosis and prognosis of cancers.

#### 4.2.3. Proteomics of iPSC-Derived Cells in Disease Models

The possibility to obtain differentiated cells from iPSC opens the opportunity to study the biology of all human cell types, including studies of genetic diseases. In a study of patients with dilated cardiomyopathy caused by mutation in the Titin (TTN) gene, skin fibroblasts from these patients and healthy donors were used to generate iPSC that were posteriorly differentiated into cardiomyocytes [127]. A mass-spectrometry analysis identified 301 unique proteins differentially expressed between dilated cardiomyopathy iPSC-derived cardiomyocytes and cardiomyocytes generated from healthy patients (151 upregulated and 150 downregulated). The downregulated proteins were associated with cardiac Ca^2+^ handling: the ryanodine receptor 2, the sarcoplasmic reticulum histidine-rich calcium-binding protein, and sodium/calcium exchanger 1 [128]. In contrast, the upregulated proteins were associated with changes in the extracellular matrix, actin-binding, and muscle proteins. These findings correlated with the clinical observations related to the disease, such as abnormal contraction and calcium handling mechanism [127].

The proteomic approaches using patient-derived iPSC have also been used to aid studies with human neurons and diseases-like AD that face some challenging experimental and practical difficulties. AD is known to promote irreversible dementia, characterized by multiple cognitive deficits [129]. Limitations associated with in vitro studies have been critical to the understanding of the disease mechanisms, like the diffusion rate of the secreted misfolded amyloid-β peptides, known to be an important cause of AD due to its deposition and accumulation [130]. At this point, we can revisit the work by Chen et al. [125], which was first considered in the preceding Section 4.2.2, to reinforce the view regarding the relevance of iPSC-based proteomic studies carried out with the use of 3-D structures. Clearly, they open the possibility of bypassing the limitations represented by the impossibility of mechanistic studies on the pathogenesis and progression of AD in tissues from living patients. Even though AD is a very complex pathology, the proteomic analysis using organoids of neurons differentiated from AD-specific iPSC derived from PBMC allowed a better understanding of molecular mechanisms underpinning the development of AD. Identification of a panel of modulated proteins instead of a single candidate, as well as of the processes and pathways in which they are involved, allowed the authors to highlight that “*proteomic profiles of our AD-derived 3D neuro-spheroid model illustrate certain levels of similarity when compared to those of human brain tissue*” [125]. As mentioned above, the proteomic profile of PBMC/iPSC-derived neurons cultivated in 3-D was compared with postmortem brain tissues, revealing similar pathological process of gliogenesis, axon development, and neuron projection development that are associated with AD. Interestingly, components associated with extracellular vesicle secretion were upregulated in the 3-D culture and the neural tissue [125], indicating that the 3-D culture model can also be used as a platform for the studies of neural cell communication. 

#### 4.2.4. Proteomic Approaches for the Study of Cell Interactions

Proteomic and iPSC generation technologies were also used to study cell-to-cell interactions. Proteomic approaches emerge as a powerful tool for understanding different aspects of cell–cell communication pathways [131]. In a model of cardiomyocyte and endothelial cells derived from human iPSC, both cells were maintained in direct coculture, and a quantitative whole proteomic approach—using sequential window acquisition of all theoretical mass spectra (SWATH-MS) [132] —was conducted to evaluate the modifications in each cell type promoted by the mutual crosstalk. The proteomic profiles of both cells were compared with the monoculture condition. In cardiomyocytes maintained in coculture, the ratio of adult to embryonic protein isoforms for troponin I (TNNI3/TNNI1), myosin heavy chain (MYH7/MYH6), and myosin light chain (MYL2/MYL7) was increased with respect to cardiomyocytes cultured separately from endothelial cells, indicating, for the first time, that cardiomyocytes acquire a more adult-like and ventricular-like phenotype due to the interaction with endothelial cells [133].

#### 4.2.5. Drug Screening Using iPSC-Derived Differentiated Cells: An Unexplored Field

iPSCs provide a valuable platform for drug/toxicity screening with more relevant and precise results obtained than with conventional animal models [134]. Several studies have differentiated iPSC into a specific mature phenotype to promote toxicity tests and screening of drugs [135]. Zhao et al. [136] performed an extensive drug screening test (approximately 1,500 compounds) to evaluate neuropsychiatric properties of the investigated drugs in the modulation of the Wnt/β-catenin pathway of human iPSC-derived neural progenitor cells, revealing a powerful strategy for the treatment of neurologic disorders.

Alterations in cellular signaling pathways by drug administration and their consequences in cell metabolism can be detected via proteomics, resulting in a sensitive method to predict the efficacy and toxicity of the treatments for different diseases. To our knowledge, no iPSC-based proteomic approaches have been applied to drug screening or toxicity tests of iPSC-derived cells. Proteomics techniques have been employed for preclinical drug discovery and toxicology using primary cultures and immortalized cells, revealing an exciting and growing field in present days (for illustrative reviews in the 2 last decades [137,138]). Therefore, the use of proteomic techniques using iPSC-derived cells—from healthy donors and patients suffering from a wide variety of diseases—may represent an important new tool in drug discovery and toxicity tests, offering the potential for more customized medicines and more accurate risk assessment as well as reduced costs and decreased use of animal models.

## 5. Conclusions

The generation of iPSC has created a wide range of possibilities in the study of cell biology and clinical applications. Such advances required techniques that can supply a great amount of information, and proteomics has been a powerful tool for this purpose. The combination of both technologies has made it possible to characterize iPSC in the early stages of reprogramming, to identify key molecules that are essential for differentiation into a more mature phenotype, and to establish a platform for testing multiple drugs for a variety of diseases, including genetic diseases, with great level of specificity. The advances in this field depend on facing the current limitations associated with both technologies. The capacity to induce a complete differentiation into a mature cell type in vitro and the complexity to mimic tissue environment interactions due to the presence of different cell types and 3-D architectural organization are some of the challenges associated with the iPSC field. In addition, the development of new approaches that can more readily screen a cell proteome in association with other omics will provide a massive quantity of information that can be challenging to store and process and will also require new bioinformatic approaches. As a result, such advances will allow the development of biological panels that will provide a wide and, at the same time, deep source of information about cellular processes associated with organ physiological and pathological processes, with important implications in clinics for optimizing treatments to specific patients.

## Figures and Tables

**Figure 1 cells-08-00703-f001:**
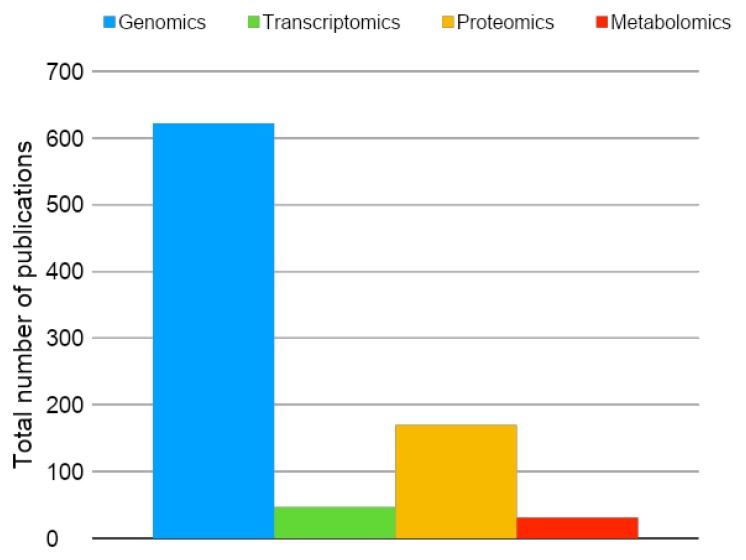
Total number of original publications (2006–2019)—available in PubMed—using omics analysis in the field of induced pluripotent stem cells (iPSC). Blue: genomics; green: transcriptomics; yellow: proteomics; red: metabolomics. Keywords for the search on May 14, 2019 were genomics, transcriptomics, proteomics, metabolomics, induced pluripotent stem cells, iPSC, and iPSCs.

**Figure 2 cells-08-00703-f002:**
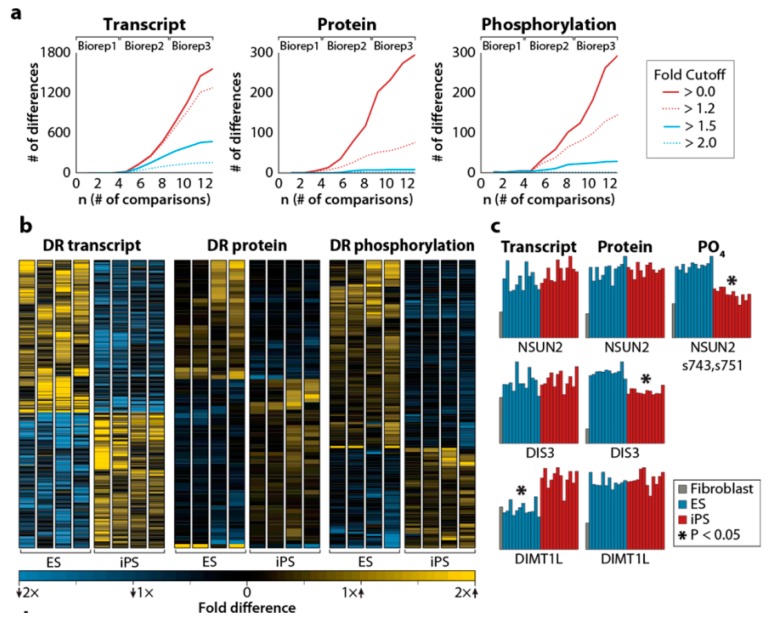
(**a**) Differentially regulated transcripts (left), proteins (center), and phosphorylation sites (right) between embryonic stem cells (ESC) and iPSC, as a function of the number of comparisons. The inset shows the fold cutoff corresponding to each colored line in the 3 panels. (**b**) Heatmap representing differentially regulated transcripts, proteins, and phosphorylation sites. Significant differences between ESC and iPSC are indicated by asterisks. (**c**) Examples of differentially regulated transcripts, proteins, and phosphorylation sites. For details, see the original article by Phanstiel et al. [61]. Reproduced from Reference [61] with permission, according to the Creative Commons Attribution 4.0 International License (https://creativecommons.org/licenses/by/4.0/).

**Figure 3 cells-08-00703-f003:**
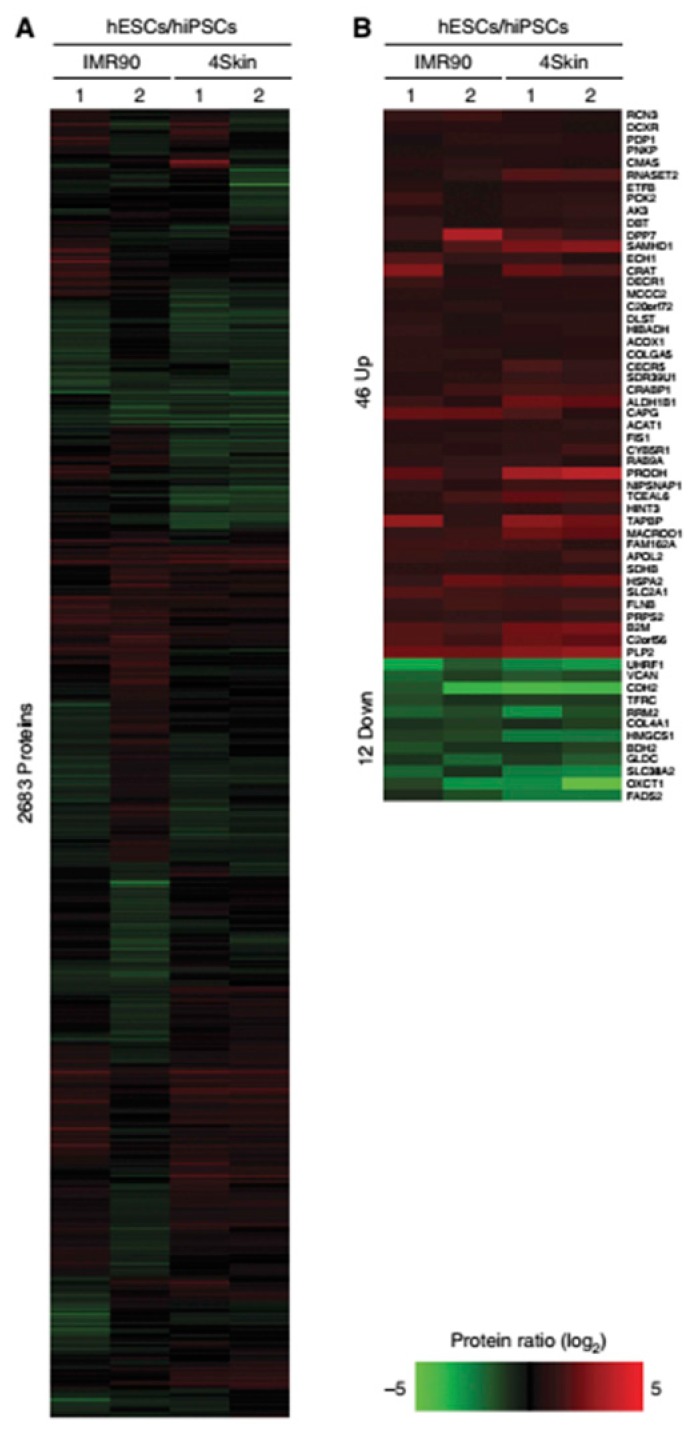
(**A**) (**A** in the original article, Reference [60]) Relative protein abundance represented as heatmaps for hESC/hiPSC (2,683 proteins). Red indicates upregulated proteins, and green indicates those that are downregulated. (**B**) (**B** in the same original article) Significance Analysis of Microarray (SAM) of 58 proteins (indicated on the right side) that were found to be significantly regulated between hESC/hiPSC. The authors compared the *log2* of hESC/hiPSC proteomes obtained in 2 different experiments: IMR90 and 4Skin. For more details, see the original article by Munoz et al. [60]. Reproduced from Reference [60] with permission, according to the Creative Commons Attribution 4.0 International License (https://creativecommons.org/licenses/by/4.0/).

**Figure 4 cells-08-00703-f004:**
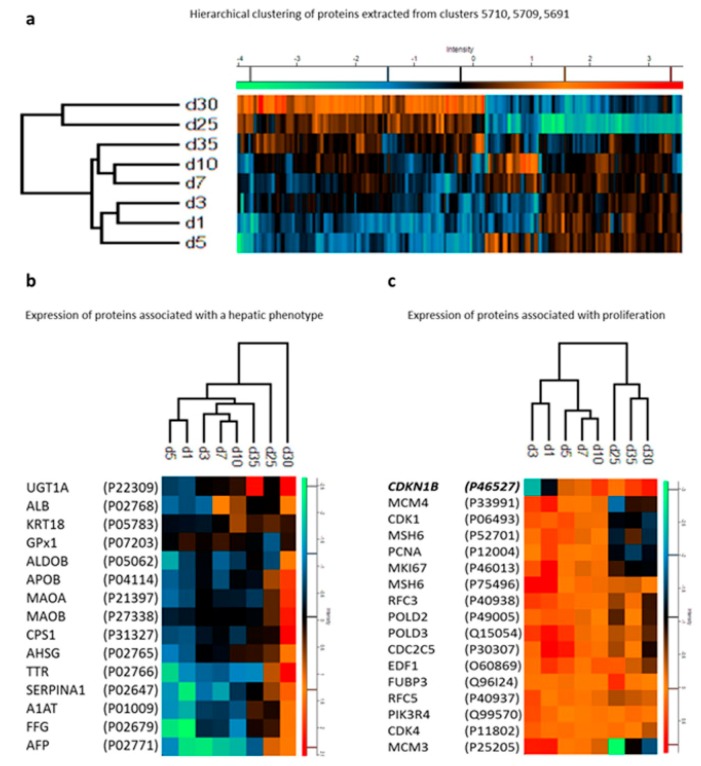
(**a**) Hierarchical clustering of proteins from hepatocyte-like cells at different differentiation stages (days). (**b**) Expression profiles of proteins marking a hepatic phenotype. (**c**) Expression profiles of proteins marking proliferation alongside cyclin-dependent kinase inhibitor 1B. For details, see the original article by Hurrell et al. [105]. Reproduced from Reference [105] with permission, according to the Creative Commons Attribution 4.0 International License (https://creativecommons.org/licenses/by/4.0/).

**Figure 5 cells-08-00703-f005:**
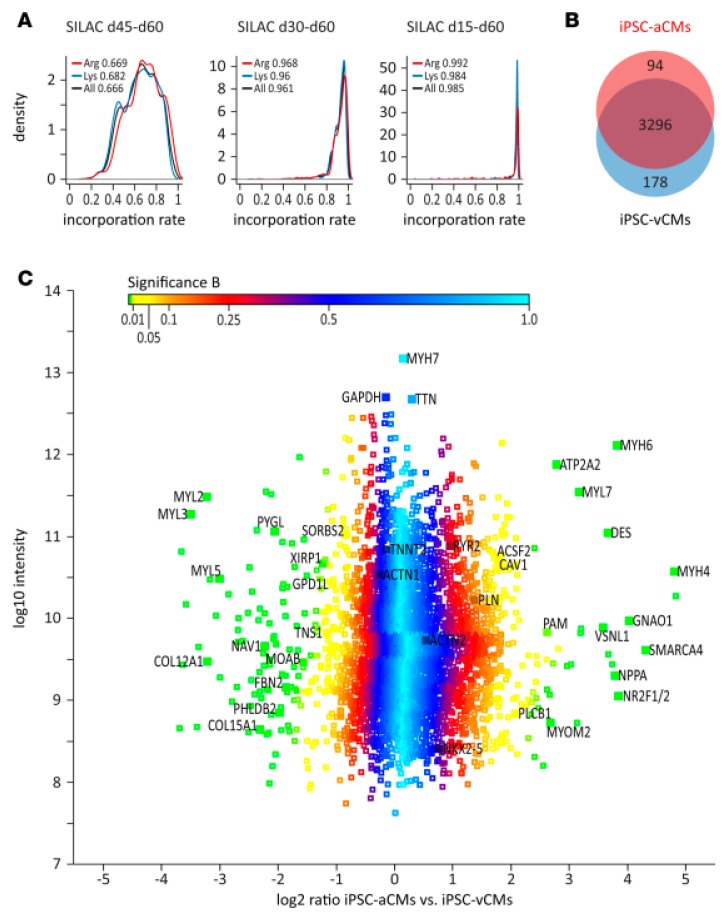
(**A**) (**A** in the original article, Reference [113]) Analysis of labelling efficiency of iPSC-derived cardiomyocytes using stable isotopes (Stable Isotope Labeling by/with Amino acids in Cell culture (SILAC)-based) after different exposure times. (**B**) (**B** in the same original article) Venn diagram representing proteins specifically and commonly expressed in atrial and ventricular cells derived from iPSC. (**C**) (**C** in the same original article) Distribution of protein ratios between atrial and ventricular cells derived from iPSC. For details, see the original article by Cyganek et al. [113]. From Reference [113] with permission, according to the Creative Commons Attribution 4.0 International License (https://creativecommons.org/licenses/by/4.0/).

**Figure 6 cells-08-00703-f006:**
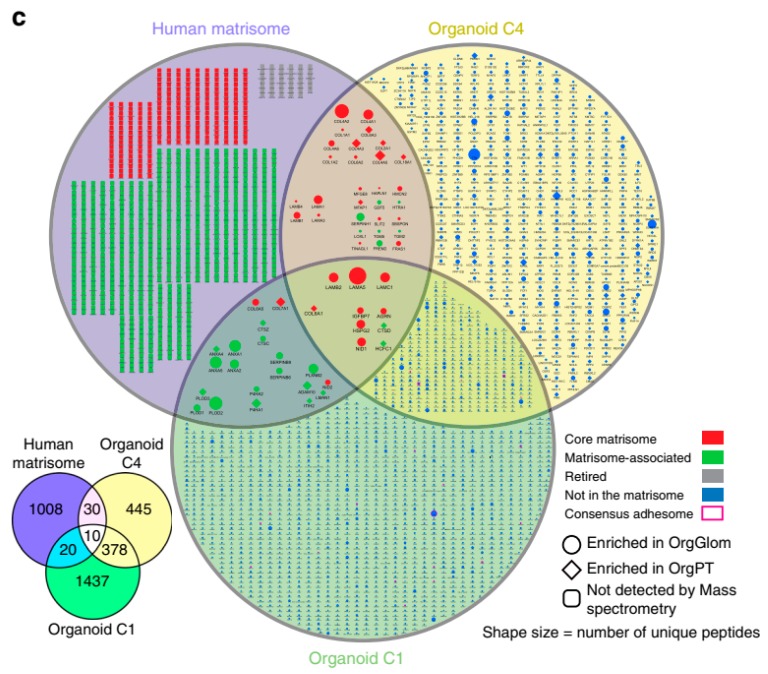
Glomeruli organoids display a maturing glomerular basement membrane matrisome. The figure corresponds to panel **4c** in the original publication, Reference [119], applying organoids proteomics data onto the human matrisome database. Sixty extracellular matrix proteins are shown in the Venn diagram (lower left). For further details regarding these and the other proteins, see the original article by Hale et al. [119]. Reproduced from Reference [119] with permission, according to the Creative Commons Attribution 4.0 International License (https://creativecommons.org/licenses/by/4.0/).

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
