# Peer review of "Proteomics in the World of Induced Pluripotent Stem Cells"

_cells, 2019, doi:10.3390/cells8070703_

Round 1

Reviewer 1 Report

The manuscript entitled "Proteomics in the World of Induced Pluripotent StemCells by Lindoso et al., is an informative and well-written review summarising the field of proteomics and iPS cells. The topic is well covered and appropriate references have been used. I would suggest that the authors may expand on the: 

4.2.2. Proteomics in iPSC-derived Organoids since this is an emerging field and as it is described here its importance in not reflecting. 

In the line 525 the expression "healthy patients" should be corrected.

Author Response

Please, see attached file.

Reviewer 2 Report

Authors provided review focusing methodological aspects of proteomics and also review basic information about iPSCs, mainly comparison of their proteome and transcriptome with embryonic stem cells. They also discuss the utilization of information about iPSCS proteome and transcriptome in modelling of diseases, studying of cell to cell interactions and drug screening. According to my opinion this review is suitable for publication in Cells.

Author Response

Please, see attached file.

Reviewer 3 Report

This review summarized a relatively comprehensive proteomics study of Induced Pluripotent Stem Cells (iPSC), giving a global view of proteomics studies history and development of iPSC. It contained most of the analysis methods and results of proteome investigation in iPSC and ESC. The content was based on wide range of publications, however, the grammar descriptions are not rigorous, Which should be carefully modified.

1.     Figures: The resolution of figures should be improved; Heatmap (Figure 3) should have a scaleplate, like 1 (the highest intensity) was represented by the deepest color of red, -1 (the lowest intensity) was illustrated by the deepest color of green.

2.     The preparation of peptides (bottom-up) in proteomics analysis is very important, different digestion methods are recommended to discussed in the review. Which is based on the “methodological aspects” in the abstract.

3.     Line 35 in Abstract “has” should be “have”.

4.     Line 49 “based in” should be “based on”.

5.     Line 80 “download” should be “downloading”.

6.     Line 190 “the first scientific questions” should be “the first scientific question”

7.     Line 265 and 266 “differently-expressed” and “differently expressed” should be consistent.

8.     Line 313, “et al” should be “et al.

9.     Line 330 “two” and “2” should be consistent

10.Line 583, delete “see”.

Author Response

Please, see attached file.
